# Regulation of Cu Species in CuO/SiO_2_ and Its Structural Evolution in Ethynylation Reaction

**DOI:** 10.3390/nano9060842

**Published:** 2019-06-01

**Authors:** Haitao Li, Lijun Ban, Zhipeng Wang, Pingfan Meng, Yin Zhang, Ruifang Wu, Yongxiang Zhao

**Affiliations:** Engineering Research Center of Ministry of Education for Fine Chemicals, School of Chemistry and Chemical Engineering, Shanxi University, Taiyuan 030006, China; banlijun1992@163.com (L.B.); zhipengw44@163.com (Z.W.); mpf15091823973@163.com (P.M.); sxuzhy@sxu.edu.cn (Y.Z.); wrf850401@sxu.edu.cn (R.W.)

**Keywords:** copper phyllosilicate, different states, formaldehyde ethynylation, 1,4-butynediol

## Abstract

A Cu-based nano-catalyst has been widely used in the ethynylation of formaldehyde; however, the effects of the presence of Cu on the reaction have not yet been reported. CuO/SiO_2_ catalysts with different Cu species were prepared by impregnation (IM), deposition–precipitation (DP), and ammonia evaporation (AE). The structural evolution of the Cu species in different states of the ethynylation reaction and the structure–activity relationship between the existence state of the Cu species and the catalytic properties of the ethynylation reaction were studied. The results show that the Cu species in the CuO/SiO_2_ (IM), prepared using the impregnation method, are in the form of bulk CuO, with large particles and no interactions with the support. The bulk CuO species are transformed into Cu^+^ with a low exposure surface at the beginning of the reaction, which is easily lost. Thus, this approach shows the lowest initial activity and poor cycle stability. A high dispersion of CuO and copper phyllosilicate exists in CuO/SiO_2_ (DP). The former makes the catalyst have the best initial activity, while the latter slows release, maintaining the stability of the catalyst. There is mainly copper phyllosilicate in CuO/SiO_2_ (AE), which is slowly transformed into a highly dispersed and stable Cu^+^ center in the in situ reaction. Although the initial activity of the catalyst is not optimal, it has the optimal service stability.

## 1. Introduction

Catalytic ethynylation of formaldehyde is an important initial chemical process for the mass production of high-value intermediates such as 1,4-butynediol (BD) and propargyl alcohol (PA) and downstream chemicals such as 1,4-butanediol (BDO), 3-butene-1-alcohol (BTO), tetrahydrofuran (THF), polytetramethylene ether glycol (PTMEG), γ-butyrolactone (GBL), poly butylenes succinate (PBS), and polybutylene terephthalate (PBT). These are in high demand for use in pharmaceuticals, textiles, the military, electrical applications, electronics, automotive manufacturing, and in other fields [1,2,3,4,5]. Stimulated and driven by the demand for downstream products, such as new materials made of PBS, the demand for 1,4-butynediol is also increasing every year [6].

The production of BD mainly adopts the Reppe method, which takes formaldehyde and acetylene as raw materials and generates a nucleophilic addition reaction by using a copper-based catalyst. From the perspective of screening practical industrial catalysts, Cu-based catalysts supported with diatomite, silica gel, or a SiO_2_–MgO complex have been reported, as well as unsupported Cu–Bi nanopowder catalysts and synthetic malachite catalysts [7,8,9,10,11,12,13,14,15]. In the ethynylation reaction, the Cu^2+^ (copper oxide or copper carbonate) precursors in the above catalysts must be transformed in situ into a cuprous acetylide active phase through a complex chemical process. This process includes the reduction of Cu^2+^ to Cu^+^ and the carbonization of Cu^+^ to produce cuprous acetylide [7]. A key problem that needs to be solved in this process is how to avoid the excessive reduction of Cu^2+^ to inactive metallic Cu, which would otherwise catalyze the formation of polyacetylene and rapidly deactivate the catalyst [7,8,9,10,11,12,13,14,15,16]. In recent years, our research group has found that the interaction of Cu species with other components can effectively inhibit the formation of inactive metal Cu and improve the stability of Cu^+^ active species. Examples of this effect include the electron-assisted action of Bi_2_O_3_, Fe_3_O_4_, and the strong interaction between CuO and a SiO_2_–MgO aerogel support [17,18]. When Cu_2_O was directly applied to the ethynylation of formaldehyde, it was still necessary to control the grain size of Cu_2_O and regulate the interaction between Cu_2_O and TiO_2_ in order to inhibit the reduction of Cu^+^ to inactive metallic Cu [19,20,21].

SiO_2_ is currently the most commonly used Cu-based catalyst support, and SiO_2_ is also commonly used as the support of the ethynylation catalyst. SiO_2_ is generally regarded as an inert support; it mainly supports and disperses the active components. However, some studies have found that regulating the introduction method of Cu species in SiO_2_ can promote the formation of a strong metal–support interaction [22,23,24,25,26]. This effectively regulates the redox properties of the Cu species [27], realizes the controllable reduction of Cu^2+^ to Cu^+^, and maintains the stable existence of Cu^+^ under the reaction condition of a H_2_ atmosphere. Based on this, it is speculated that maintaining the stability of Cu^+^ in a hydrogen atmosphere by regulating the existing state of the copper species and the interaction between the copper and silicon oxide can also be applied to the reductive atmosphere of formaldehyde ethynylation. However, no literature about the effects of copper species regulation on formaldehyde ethynylation performance for CuO/SiO_2_ catalysts has been reported.

Therefore, a series of CuO/SiO_2_ catalysts with different Cu species were prepared. Preparation techniques include impregnation (IM), deposition–precipitation (DP), and ammonia evaporation (AE). The structural evolution and catalytic performance of the copper species in formaldehyde acetylation were studied.

## 2. Materials and Methods

### 2.1. Preparation of CuO/SiO_2_ Catalysts

The CuO/SiO_2_ catalysts were prepared by impregnation, deposition–precipitation, and ammonia evaporation methods, all with SiO_2_ as the support. The nominal Cu loading of the three catalysts was 20 wt%. In the following subsections, we describe the details of each method.

#### 2.1.1. Impregnation

A total of 3.8 g of Cu(NO_3_)_2_·3H_2_O was dissolved in 60 mL of deionized water at ambient temperatures to obtain an aqueous solution. Then, 3.75 g of nano-SiO_2_ powder was added to form a suspension. After stirring for 8 h, the suspension was dried at 110 °C overnight. The final sample was calcined at 450 °C for 4 h. The catalyst prepared by this method is denoted as CuO/SiO_2_(IM).

#### 2.1.2. Deposition and Precipitation

A total of 3.8 g of Cu(NO_3_)_2_·3H_2_O was dissolved in 112 mL of deionized water at ambient temperature to obtain an aqueous solution. Then, 3.75 g of nano-SiO_2_ powder was added to form a suspension. A total of 112 mL of Na_2_CO_3_ aqueous solution was added dropwise to the suspension under stirring at ambient temperatures. After keeping the mixture at ambient temperature for 2 h, the mixture was filtered, washed with deionized water, dried at 110 °C overnight, and calcined at 450 °C for 4 h. The catalyst prepared by this method is denoted as CuO/SiO_2_(DP).

#### 2.1.3. Ammonia Evaporation

A total of 3.8 g of Cu(NO_3_)_2_·3H_2_O was dissolved in 40 mL of deionized water, and 25% ammonia aqueous solution was then added to it. The mixed solution was stirred for 30 min to form a copper ammonia complex solution with a pH value of 10. A total of 11.4 g of nano-SiO_2_ powder was subsequently added into the copper ammonia solution and stirred for another 30 min. All of this was done at room temperature. Then, the suspension was heated at 70 °C to evaporate the ammonia and deposit the copper species on the nano-SiO_2_ powder. The evaporation process was not stopped until the pH value of the suspension decreased to a value between 6 and 7. Then, the precipitate acquired from centrifugation was washed with deionized water three times and dried at 80 °C for 10 h. The catalyst precursor was calcined in air at 450 °C for 4 h. The catalyst prepared by this method is denoted as CuO/SiO_2_(AE).

### 2.2. Characterization of CuO/SiO_2_ Catalysts

N_2_-physisorption analyses were performed using a Micromeritics ASAP-2020 apparatus (Norcross, Georgia (GA), USA). The samples were degassed at 150 °C for 5 h before the test. Specific surface areas were obtained using the multi-point Barrett-Emmett-Teller (BET) method. The pore-size distribution was determined by the Barrett–Joyner–Halenda (BJH) method from the desorption branches of the adsorption isotherms.

X-ray diffraction (XRD) patterns of the samples were recorded with a D8 Advance diffractometer (Bruker Corporation, Billerica, MA, USA) with Cu Kα radiation (λ = 1.5418 Å).

Fourier-transform infrared (FTIR) spectra were recorded with a Nicolet™ iS50 spectrophotometer (Thermo Fisher Scientific, Waltham, MA, USA) in the range of 400–4000 cm^−1^.

Transmission electron microscopy (TEM) experiments were performed using a JEOL JEM-2100 transmission electron microscope operating at 200 kV (JEOL, Tokyo, Japan). TEM samples were dispersed by sonication in ethanol, followed by deposition of the suspension onto a standard Cu grid covered with a porous carbon film.

Raman spectroscopy was performed with a LabRAM HR Evolution Raman spectrograph (HORIBA Scientific, Paris, France) and a 532 nm laser operated at 0.08 mW.

A Thermal gravity-Differential thermal gravity (TG-DTG) test of the sample was carried out on a NETZSCH STA449C thermal analysis instrument. Samples were measured after a stent under a 20 mL/min N_2_ atmosphere, with a temperature increase of 10 °C/min up to 800 °C.

X-ray photoelectron spectroscopic (XPS) measurements were conducted on an ESCALAB 250 spectrometer (Thermo Fisher Scientific, Waltham, MA, USA) using an Al Ka X-ray source (hν = 1486.7 eV).

X-ray auger electron spectroscopy (XAES) was performed on a PHI 1600 ESCA spectrometer (Perkin-Elmer, Waltham, MA, USA) equipped with a monochromatic Al Ka X-ray source (hm = 1361 eV) operating at a pass energy of 100 eV.

CO-IR spectra were recorded on a Bruker Tensor 27 Fourier-transform infrared spectrometer (Bruker, Billerica, MA, USA) equipped with a highly sensitive MCT detector cooled using liquid N_2_. The used samples were evacuated at 60 °C for 3 h. The temperature of the sample was reduced to 25 °C and then CO was adsorbed for 1 h. After saturation, the sample was treated under a vacuum for 10 min. The IR spectra were then recorded.

The Cu content in the reaction solution was measured by means of inductively-coupled plasma (ICP) spectroscopy with iCAP 7400 ICP-OES (Thermo Fisher Scientific, Waltham, MA, USA) equipment.

### 2.3. Catalysis Tests of the CuO/SiO_2_ Catalysts

Evaluation of the catalysts was carried out in a three-neck flask connected to a reflux condenser. A certain amount of catalyst and a 50 mL formaldehyde (35 wt%) water solution were mixed using electromagnetic stirring in a flask placed in an oil bath. A flow of pure N_2_ was introduced into the flask to purge the O_2_. Then, the solution with the catalyst was heated to the reaction temperature (90 °C) under continuous stirring conditions. Subsequently, a C_2_H_2_ stream was switched on to start the ethynylation reaction. Several hours later, the catalytic reaction was stopped by decreasing the reaction temperature to room temperature, introducing the N_2_ stream, and closing the C_2_H_2_ stream. Cyclic experiments were carried out to determine the stability of the catalyst. For each cycle, the used catalyst running for 10 h was separated with the reaction solution and washed with deionized water, and then a new 10 h run was started.

The used catalyst was centrifugated and washed with deionized water and dried under a vacuum at ambient temperature. In the centrifugate, the formaldehyde content was determined by titration with sodium thiosulfite, while the 1,4-butynediol content was analyzed using an Agilent 7890 A gas chromatography setup with a 1,4-butanediol-added internal standard method [28]. The selectivity of 1,4-butynediol was obtained by dividing the yield of 1,4-butynediol by the conversion of formaldehyde.

## 3. Results

### 3.1. Textural Properties of CuO/SiO_2_ Catalysts

The adsorption–desorption isotherm and the pore size distribution curve of each catalyst and SiO_2_ are illustrated in Figure 1. The textural properties of each sample are summarized in Table 1. It can be seen from Figure 1a that each of the samples exhibits Langmuir type IV isotherms with an H2-type hysteresis loop, indicating that a mesoporous structure is present in all cases. However, the P/P_0_ ranges of the hysteresis loops are significantly different, indicating that the pore distributions of the respective samples are different. The specific surface area of SiO_2_ is 328 m^2^·g^–1^, and the pore size is distributed in the range of 15–60 nm. Among the three catalysts, CuO/SiO_2_(IM) has the smallest specific surface area and the largest average pore size and pore volume. Its pore size is distributed in the range of 10–40 nm. The average pore size and pore volume of CuO/SiO_2_(IM) are even larger than that of SiO_2_, which be presumed that this is due to the accumulation of large CuO particles. CuO/SiO_2_(AE) shows the largest specific surface area and the smallest average pore size and pore volume. Its pore size distribution is bimodal, and the most probable pore sizes are 2.5 and 8 nm. The specific surface area, pore size, and pore volume of CuO/SiO_2_(DP) are between those of CuO/SiO_2_(IM) and CuO/SiO_2_(AE), and the most probable pore sizes are 2.5 and 11 nm. The CuO/SiO_2_(DP) and CuO/SiO_2_(AE) samples exhibit a bimodal pore size distribution, indicating that there is a strong interaction between the Cu species and the support in the sample, which causes the destruction of the support structure to create new pores [29].

### 3.2. Structural Analysis of CuO/SiO_2_ Catalysts

Figure 2 shows the XRD patterns of the catalysts. It is evident that, in addition to the dispersed peak of amorphous SiO_2_ observed at the 2*θ* value of 22.5°, there were different forms of Cu species in all of the CuO/SiO_2_ catalysts before and after calcination [30]. The CuO/SiO_2_(IM) sample before calcination exhibits strong and sharp characteristic peaks at 2*θ* values of 12.9°, 25.8°, and 36.5° (Figure 2a), which is attributed to the crystallized Cu_2_(OH)_3_NO_3_ species (JCPDS No. 75-1779). After calcination (Figure 2b), the diffraction peaks of the crystal phase for CuO species appear at 2*θ* values of 32.2°, 35.2°, 38.4°, 48.4°, 53.2°, 58.0°, 61.2°, 65.9°, and 67.7° (JCPDS No. 044-0706) [31]. The peak shape is sharp, indicating that Cu_2_(OH)_3_NO_3_ decomposed to form large CuO particles. For the CuO/SiO_2_(DP) sample before calcination, weak diffraction peaks of Cu_2_CO_3_(OH)_2_ are observed at 2*θ* values of 14.8°, 17.6°, 24.1°, 31.2°, and 35.6° (JCPDS No. 41-1390). The calcined sample shows weaker diffraction peaks of CuO at 2*θ* values of 35.2°, 38.4°, and 48.4°, indicating that the dispersion of the CuO species was better compared to that of the CuO/SiO_2_(IM) sample. The CuO grain sizes in CuO/SiO_2_(IM) and CuO/SiO_2_(DP) are 16.2 and 7.6 nm (Table 1), respectively. Compared with the diffraction peak area of CuO, it is evident that the area of CuO/SiO_2_(DP) is significantly lower than that of CuO/SiO_2_(IM). Under the premise that the Cu content of the two samples is similar, it is speculated that there were more dispersed Cu species in CuO/SiO_2_(DP), which were below the XRD detection limit.

For CuO/SiO_2_(AE), no obvious diffraction peaks of Cu species are observed before and after calcination. In the sample after calcination, only very weak diffraction peaks are shown at 2*θ* = 31.2 and 35.8°, which are attributed to the characteristic peak of copper silicate [32]. This is because, during the preparation of the catalyst using the ammonia evaporation method, cuprammonia complexes react with the silica to form a phyllosilicate. According to the literature [33,34], in the CuO/SiO_2_(AE) catalyst, the Cu species contains CuO and phyllosilicate. The appearance of CuO is due to the partial decomposition of phyllosilicate after calcination at 450 °C, which obtains a well-dispersed CuO. Another part of CuO may be produced by the degradation of Cu(OH)_2_ formed by the hydrolysis of cuprammonia complexes after the pH value decreased during ammonia evaporation. However, this part of CuO species cannot be detected in the XRD patterns, indicating that the CuO in the sample prepared using the ammonia evaporation method is highly dispersed or amorphous.

### 3.3. Morphological Analysis

Figure 3 shows the TEM image of each catalyst after calcination. It is evident that the morphology of CuO/SiO_2_(AE) is different from that of the others. CuO/SiO_2_(IM) (Figure 3a) and CuO/SiO_2_(DP) (Figure 3b) show a large number of spherical CuO particles. However, CuO/SiO_2_(AE) (Figure 3c) presents a layered morphology belonging to the Cu-layered silicate phase.

### 3.4. FTIR Analysis

Figure 4a shows the IR spectra of the CuO/SiO_2_ catalysts before calcination. The symmetric stretching vibrational peaks of the Si–O bonds in amorphous SiO_2_ are observed at 470 and 800 cm^−1^, and the asymmetric stretching vibrational peak of the Si–O–Si bond is observed at 1100 cm^−1^ [35,36]. The bending vibrational peak of the O–H bond for water appears at around 1630 cm^−1^ [37]. At 1388 cm^−1^, the asymmetric stretching vibrational peak of (νNO_3_) appears. Additionally, the asymmetric stretching vibrational peaks of the NO_3_ group are observed at 1337 and 1427 cm^−1^ in the CuO/SiO_2_(IM) sample, which further confirms the existence of the Cu_2_(OH)_3_(NO_3_) species as observed in the XRD patterns [38]. The characteristic peak of (CO_3_)^2−^ appears at 1500 cm^−1^ in CuO/SiO_2_(DP), indicating the presence of Cu_2_CO_3_(OH)_2_ or CuCO_3_ species. This is consistent with the XRD results. The infrared absorption peak at 1040 cm^−1^ is also observed in the CuO/SiO_2_(DP) and CuO/SiO_2_(AE) samples, which are derived from the red shift of infrared absorption at 1100 cm^−1^ in SiO_2_. This indicates the presence of an Si–O–Cu bond.

After calcination at 450 °C, it is found that the absorption peaks belonging to (NO_3_)^−^, (CO_3_)^2−^, H_2_O, and other anionic groups disappear in each sample (Figure 4b). This is due to the removal of the physical adsorption and the crystal water, and due to the decomposition of basic salt, basic nitrate, basic carbonate, or the carbonate of Cu after calcination. The shoulder peak at 568 cm^−1^ indicates the formation of CuO species in the CuO/SiO_2_ after calcination [39,40]. The absorption peaks at 1040 and 670 cm^−1^ are more obvious in the CuO/SiO_2_(DP) and CuO/SiO_2_(AE) catalysts after calcination, indicating the presence of copper phyllosilicate [41].

Combined with the XRD results, it is inferred that CuO/SiO_2_(IM), the catalyst prepared using the impregnation method, produced large-grained Cu_2_(OH)_3_(NO_3_) after drying, which transformed into large-grained CuO with a weak interaction with SiO_2_ after calcination. Under the basic conditions, Cu_2_CO_3_(OH)_2_ species were formed in CuO/SiO_2_(DP) and converted into highly dispersed CuO after calcination. Meanwhile, Cu and SiO_2_ were bonded to form an Si–O–Cu bond, resulting in a strong interaction. Under the strong basic conditions in CuO/SiO_2_(AE), Cu was completely bonded to SiO_2_ to form Si–O–Cu units, which exist in the form of copper phyllosilicate.

### 3.5. TG Analysis

Figure 5 illustrates the TG-DTG curves of the CuO/SiO_2_ catalysts. It is evident that there are two obvious mass loss steps for the CuO/SiO_2_(IM) sample. The mass loss occurring in the range of 30–150 °C is attributed to the removal of physisorbed water. The mass loss occurring between 210 and 300 °C is associated with the decomposition of Cu_2_(OH)_3_(NO_3_) into CuO. Multiple mass loss steps appear in the CuO/SiO_2_(DP) sample. Combined with the XRD and FTIR characterizations, it is speculated that the mass loss step between temperatures 30 and 155 °C is likely associated with the removal of physisorbed water. Meanwhile, the broad mass loss step in the range of 220–500 °C is possibly associated with the decomposition of Cu(OH)_2_, Cu_2_(OH)_2_(NO_3_), and Cu_2_CO_3_(OH)_2_ into CuO. For the CuO/SiO_2_(AE) sample, only one mass loss step appears. This significant mass loss step appears at 30–120 °C, which is attributed to the physical adsorption of water and NH_3_ removal on the catalyst surface. The slow decrease between 100 and 700 °C indicates that there is a small amount of decomposition of copper phyllosilicate.

### 3.6. Raman Spectra Analysis

To further verify the existence and chemical environment of the Cu species on the catalyst surface, Raman characterization was carried out. CuO is a monoclinic structure with a space group symmetry of C_2h_^6^. According to space group theory, there are 12 zone-center optical-phonon modes, Γ = 4A_u_ + 5B_u_ + A_g_ + 2B_g_, three of which (A_u_ + 2B_u_) are acoustic modes, six of which (3A_u_ + 3B_u_) are infrared active modes, and three of which (A_g_ + 2B_g_) are Raman active modes. It can be seen from Figure 6 that four distinct Raman peaks are observed for CuO/SiO_2_(IM) and CuO/SiO_2_(DP). Three of these peaks—those at 290, 342, and 625 cm^−1^—correspond to the A_g_, B_g_, and B_g_ modes of CuO crystals, respectively. These are consistent with the Raman vibrational spectrum of the CuO single crystal reported in the literature [42,43,44,45], indicating that the structure of bulk CuO exists in both samples. According to the Raman peak at 1119 cm^−1^, it is presumed that due to the weak interaction between CuO and SiO_2_, crystal defects cause a non-Raman active vibration to occur. Comparing the peak intensities of the two samples, it is found that the peak intensity of CuO/SiO_2_(DP) is significantly lower than that of CuO/SiO_2_(IM). This can be attributed to the fact that the CuO is more dispersed and the CuO particle size is smaller in CuO/SiO_2_(DP). In the CuO/SiO_2_(AE) catalyst, only a weak Raman vibrational peak is observed at 265 cm^−1^, and the presence of this peak is also observed in CuO/SiO_2_(DP). This peak has obvious displacement and broadening compared with the Raman peak at 290 cm^−1^ in bulk CuO. This indicates the presence of Cu species with different chemical environments compared to CuO, which is presumed to be copper phyllosilicate.

### 3.7. XPS Characterization

XPS was carried out to determine the chemical state of copper species, as well as their compositions. The XPS spectra of Cu 2p is shown in Figure 7. It is evident that there is only a peak at 933.5 eV in CuO/SiO_2_(IM), which is attributed to Cu^2+^ in CuO [46]. In addition to the peak at 933.5 eV, in CuO/SiO_2_(AE) and CuO/SiO_2_(DP) a significant XPS peak appears at a higher binding energy of 936.0 eV, which is attributed to Cu^2+^ in the copper phyllosilicate species [47,48,49,50]. An Si–O–Cu structure is formed in the copper phyllosilicate. Since the electrons of the Cu^2+^ outer layer are attracted by the Si–O bond, the electronic density decreases and the binding energy increases [51]. This also indicates that there is a strong interaction between Cu and SiO_2_ in the copper phyllosilicate.

Based on the characterizations of XRD, IR, Raman, etc., it can be inferred that Cu in CuO/SiO_2_(IM) exists only in large copper oxide particles and has a weak interaction with the supports. The Cu species in CuO/SiO_2_(DP) are complex. There are CuO particles with good dispersion and no interactions with the supports, and copper silicate is also present. A small amount of highly dispersed CuO is present in CuO/SiO_2_(AE), as well as a large amount of copper phyllosilicate. Therefore, there are strong interactions in CuO/SiO_2_(DP) and CuO/SiO_2_(AE), which can prevent CuO species from aggregating and growing. This is the reason why CuO/SiO_2_(DP) and CuO/SiO_2_(AE) exhibit a higher dispersion of Cu species in the XRD.

### 3.8. X-Ray Auger Spectra Analysis of the Used Catalysts

In the ethynylation of formaldehyde, the catalytic activity was shown after in situ activation of the initial CuO species, and the characterization results of the used catalyst are directly related to its catalytic performance. Figure 8 shows the Cu LMM X-ray auger electron spectra (XAES) of the used catalysts. Two characteristic peaks are observed at 915.5 and 917.5 eV. Among them, the kinetic energy at 917.5 eV is derived from the characteristic absorption of Cu^2+^ [52]. The kinetic energy at 915.5 eV is significantly lower than that of Cu^+^ in Cu_2_O at 916.5 eV, and the Cu^+^ is presumed to be Cu^+^ in the cuprous acetylide species. Since the electronegativity of C is smaller than that of O and the density of the electron around Cu^+^ in the cuprous acetylide is higher than that in Cu_2_O, the Cu^+^ is shifted to a lower binding energy compared with Cu_2_O.

For the catalysts after two cycles (Figure 8a), the only characteristic peak is at 915.5 eV, which can be attributed to Cu^+^ in cuprous acetylide. This is observed in CuO/SiO_2_(IM). Beside the peak at 915.5 eV, the characteristic peak at 917.7 eV, which belongs to Cu^2+^, is also observed in CuO/SiO_2_(DP) and CuO/SiO_2_(AE). After six cycles (Figure 8b), the characteristic peak at 917.7 eV disappears in CuO/SiO_2_(DP). Meanwhile, the 917.7 eV peak decreases in CuO/SiO_2_(AE). These results indicate that CuO in CuO/SiO_2_(IM) can be rapidly converted to cuprous acetylide due to the weak interaction between CuO and SiO_2_. In CuO/SiO_2_(DP) and especially in CuO/SiO_2_(AE), the conversion rate of Cu^2+^ to cuprous acetylide is slower. Combined with the pre-characterization results of the catalyst, CuO/SiO_2_(AE) is dominated by copper phyllosilicate, and a Cu–O–Si bond is formed between the Cu and SiO_2_ supports. This greatly inhibits the reduction reaction in the reductive atmosphere of the ethynylation of formaldehyde. In the ethynylation of formaldehyde, only a part of Cu^2+^ is converted into cuprous acetylide, thereby functioning as an active center of the ethynylation reaction. During the subsequent reaction, Cu^2+^ is slowly converted into an active cuprous acetylide species.

### 3.9. CO-IR Spectra of the Used Catalysts

To further determine the relative concentration of Cu species on the surface of each catalyst after the reaction, CO-IR spectra were recorded and are shown in Figure 9. For the Cu^n+^–CO system, the typical vibrational peaks of CO molecules at the Cu^0^, Cu^+^, and Cu^2+^ sites correspond to 2100–2000, 2140–2100, and 2150–2190 cm^−1^ in the infrared band, respectively [53,54,55,56]. It can be seen that the infrared vibrational peak at about 2124 cm^−1^ belongs to the CO molecule at the Cu^+^ site, and the infrared vibrational peak at about 2169 cm^−1^ belongs to the CO molecule at the Cu^2+^ site. The adsorbed CO on Cu^+^ is only observed in CuO/SiO_2_(IM), and the peak area decreases as the cycle times increase. This indicates that Cu^2+^ is quickly converted into the Cu^+^ center during the initial stage of the reaction, and the exposed Cu^+^ sites decrease as the reaction times increase. It can be presumed that the decrease of the exposed Cu^+^ is due to the loss of cuprous acetylide. In the initial use of CuO/SiO_2_(DP) and CuO/SiO_2_(AE), in addition to the adsorbed CO on Cu^+^, some absorbed CO on Cu^2+^ is observed. As the cycle number increases, the CO adsorption peak corresponding to Cu^2+^ decreases, while the CO adsorption peak corresponding to Cu^+^ increases. These results indicate that, during the initial stage of the reaction, only the CuO having weak interactions with the supports are reduced to Cu^+^ and converted into cuprous acetylide. As the reaction progresses, the stable Cu^2+^ in the Si–O–Cu structure is slowly converted into the cuprous acetylide species, thereby increasing the Cu^+^ center. Cu^+^ converted from the Si–O–Cu structure maintains good stability in the ethynylation of formaldehyde. Cu^+^ is not excessively reduced to metallic Cu. It is not lost, which is beneficial to the stability of the catalyst. This is consistent with the XAES results. Also, the ICP analysis confirm the results. Table 2 lists the content of Cu in liquid samples after the cyclic experiment. As can be seen, the Cu loss of the CuO/SiO_2_(IM) sample is the largest, followed by CuO/SiO_2_(DP), and the loss of the CuO/SiO_2_(AE) sample is the smallest.

### 3.10. Catalytic Performance

The reaction process for the ethynylation of formaldehyde can be described as follows:HC ≡ CH + HCHO→HC ≡ CCH_2_–OH→HO–CH_2_C ≡ CCH_2_–OH.

An excess of acetylene is employed in the ethynylation of formaldehyde reaction. Thus, the catalytic performance was characterized by the conversion of formaldehyde and the yield of the target product BD. Part of the conversion of formaldehyde was used in Cu^2+^ to Cu^+^ reductants. As reactants, the possible products are propargyl alcohol and BD. In this study, only BD was observed in the reaction materials, and no byproducts, such as proparinol, appeared. The variation of BD yield in relation to reaction time is shown in Figure 10. It can be seen that almost no BD is detected for any of the CuO/SiO_2_ catalysts during an induction period of 1 h. The presence of the induction period may be correlated with the phase evolution of the catalysts. After the induction period, the yield of BD increases with time for all CuO/SiO_2_ catalysts. At 10 h, the yield of BD for CuO/SiO_2_(DP) reaches a maximum of 87.4%. CuO/SiO_2_(AE) shows lower activity than CuO/SiO_2_(DP), with a BD yield of approximately 66.7% at 10 h. On the other hand, the BD yield of CuO/SiO_2_(IM) is only about 50.4% at 10 h.

From Figure 10b, it is evident that the initial activity and stability of the CuO/SiO_2_(IM) samples are the lowest. The yield of BD decreases rapidly from 50.4 to 20% during the 10-cycle experiments. The CuO/SiO_2_(DP) sample shows the highest initial activity, and the BD yield reaches about 87.4% in the first cycle. The yield of BD drops rapidly to 59.6% in the second cycle, and the rate of decline slows down as the reaction times are further increased. The yield of BD decreases to about 50.6% after 10 cycle reactions. For CuO/SiO_2_(AE), the yield of BD increases from 66.7 to 82.6% after two cycle reactions. After 10 cycles, the yield of BD only decreases to approximately 75%.

## 4. Discussion

In the ethynylation of formaldehyde, catalytic activity is proportional to the Cu^+^ amount. Combined with the characterization of the catalysts, it was found that there are only CuO species with large grains and weak interactions with the supports in CuO/SiO_2_(IM) (Figure 2, Figure 4, and Figure 7). The CuO species can be rapidly converted into cuprous acetylide active species. Just one reaction cycle is needed and all CuO is converted into cuprous acetylide. However, due to the large grain size of the initial CuO, the converted cuprous acetylide exposed very few Cu^+^ centers (Figure 8 and Figure 9), which caused the low initial activity of CuO/SiO_2_(IM) (Figure 10). At the same time, the weak metal–support interaction also led to the rapid loss of the active species of cuprous acetylide in the reaction process (Figure 8 and Figure 9), which further made the catalyst show poor stability. This process is clearly illustrated in Figure 11b.

There were two Cu species in CuO/SiO_2_(DP): the CuO species with high dispersion and weak interactions with the support, and the copper phyllosilicate species that have strong interactions with the support (Figure 2, Figure 4, and Figure 7). The former can be rapidly transformed into highly dispersed cuprous acetylide species in the reaction, exposing a large number of Cu^+^ active centers (Figure 8 and Figure 9), so that the catalyst has a high initial activity (Figure 10). Similar to CuO/SiO_2_(IM), this species is also prone to loss, resulting in a significant decline in catalytic activity. In copper phyllosilicate species, the Cu^2+^ slowly transformed into Cu^+^, and then slowly transformed into cuprous acetylide active species, which played a catalytic role. Due to the strong interaction between Cu species and the support in copper phyllosilicate, the resulting Cu^+^ had good stability, inhibiting the loss of the Cu^+^ active species in the reaction process. Thus, it showed high catalytic activity and service stability in subsequent reactions. This process is clearly illustrated in Figure 11a.

As mentioned above, there were only a few dispersed CuO species in CuO/SiO_2_(AE), which were initially transformed into less Cu^+^, thus showing a low initial activity. As the reaction progressed, the stable copper phyllosilicate was slowly transformed into the activity center of cuprous acetylide, so that the catalyst showed the best activity and service stability.

## 5. Conclusions

The state of Cu species in a CuO/SiO_2_ catalyst was regulated by changing the preparation method. The structure–activity relationship between the Cu species and its catalytic properties for the ethynylation of formaldehyde was studied. Studies have shown that CuO species with large particles and no interaction with supports can be rapidly transformed into the active cuprous acetylide species. However, low dispersion leads to less exposure of Cu^+^ active sites, which is manifested as low ethynylation performance. At the same time, Cu species are easily lost, resulting in a rapid loss of catalysts. CuO species with high dispersion and no interactions with the support can be rapidly transformed into the highly dispersed cuprous acetylide active species, thus presenting a high initial activity of the catalyst. However, there is also catalyst deactivation caused by the loss of active components. Among the Cu species of copper phyllosilicate, the rate of conversion from Cu species to the active Cu^+^ center was slow, and the initial activity was not ideal. However, Cu^2+^ was slowly transformed into Cu^+^ in the subsequent reaction, which improved the catalyst’s ethynylation performance. Moreover, the active species showed good anti-loss performance, and the catalyst had the best service stability.

## Figures and Tables

**Figure 1 nanomaterials-09-00842-f001:**
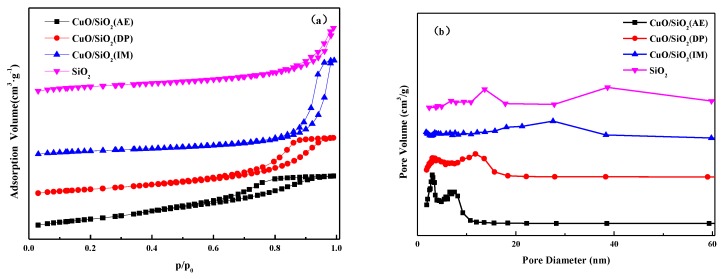
(**a**) N_2_ adsorption–desorption isotherms and (**b**) pore size distribution curves of CuO/SiO_2_ catalysts.

**Figure 2 nanomaterials-09-00842-f002:**
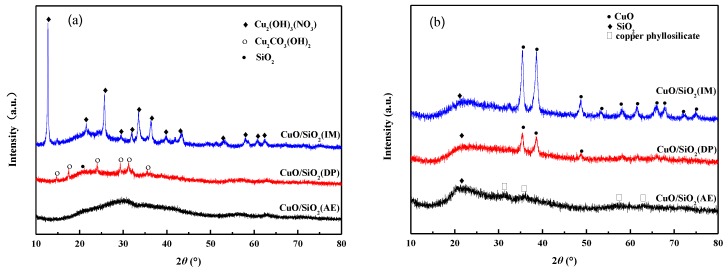
X-ray diffraction (XRD) patterns of CuO/SiO_2_ catalysts (**a**) before and (**b**) after calcination.

**Figure 3 nanomaterials-09-00842-f003:**
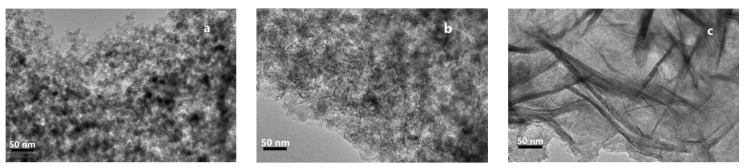
TEM images of (**a**) CuO/SiO_2_(IM), (**b**) CuO/SiO_2_(DP), and (**c**) CuO/SiO_2_(AE).

**Figure 4 nanomaterials-09-00842-f004:**
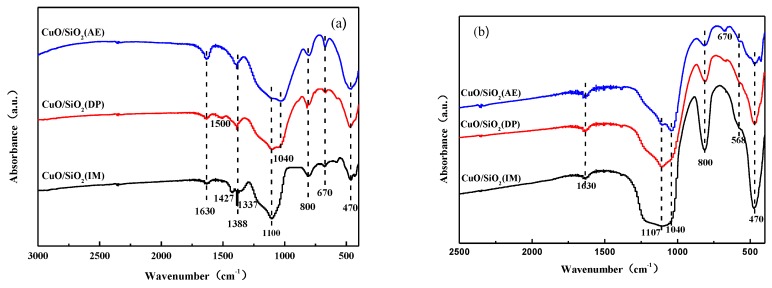
FTIR spectra of CuO/SiO_2_ catalysts (**a**) before and (**b**) after calcination.

**Figure 5 nanomaterials-09-00842-f005:**
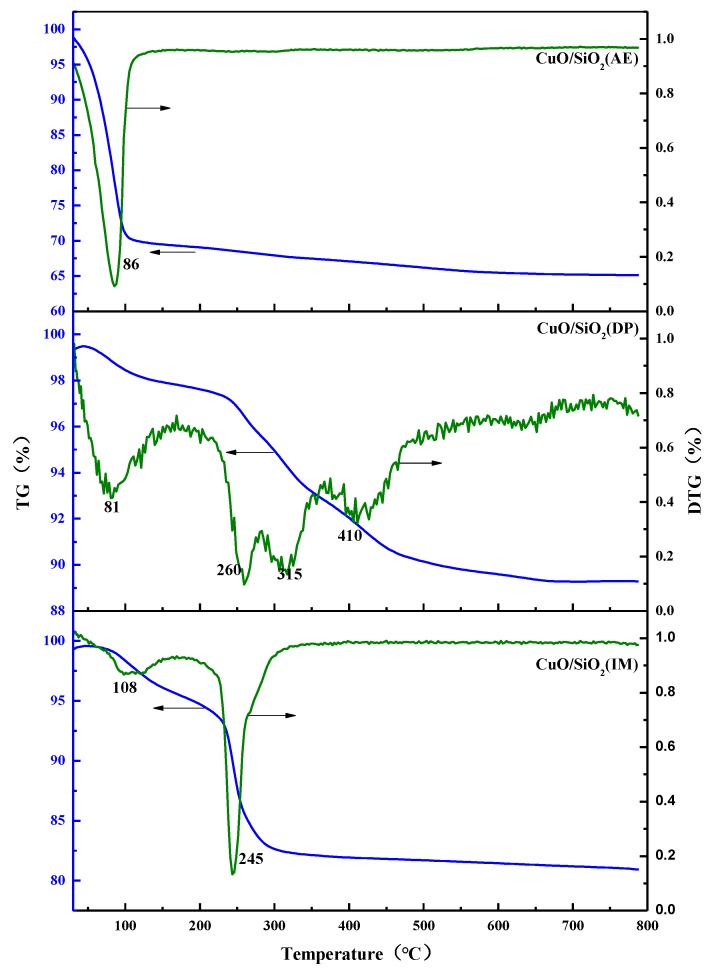
TG-DTG profiles of CuO/SiO_2_ catalysts.

**Figure 6 nanomaterials-09-00842-f006:**
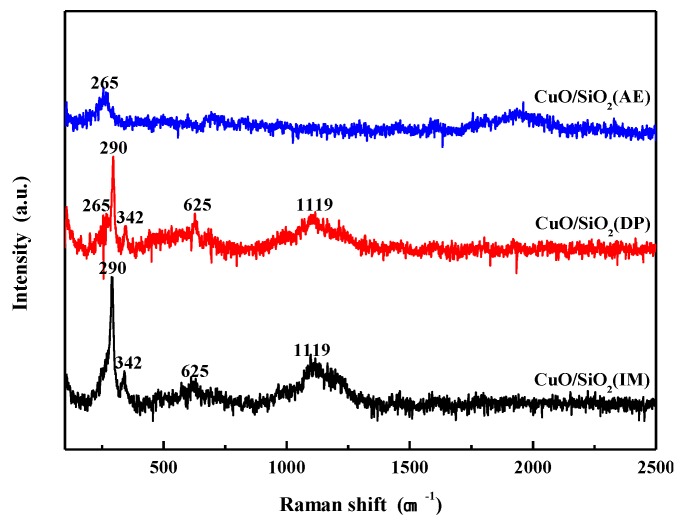
Raman spectra of CuO/SiO_2_ catalysts.

**Figure 7 nanomaterials-09-00842-f007:**
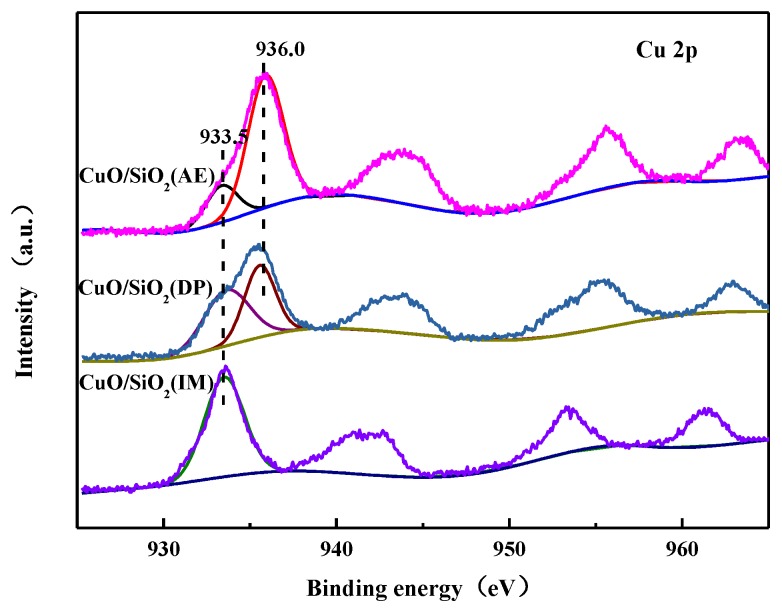
X-ray photoelectron spectra (XPS) of CuO/SiO_2_ catalysts.

**Figure 8 nanomaterials-09-00842-f008:**
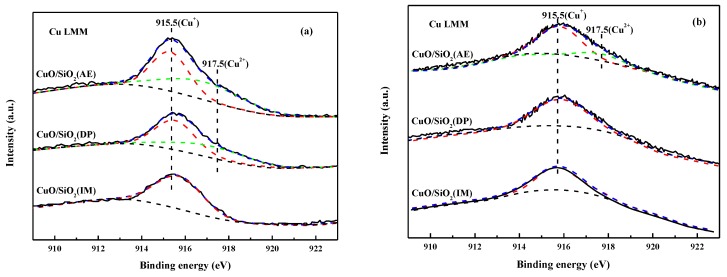
Cu LMM X-ray auger electron spectra (XAES) of CuO/SiO_2_ catalysts used after (**a**) two cyclic reactions and (**b**) six cyclic reactions.

**Figure 9 nanomaterials-09-00842-f009:**
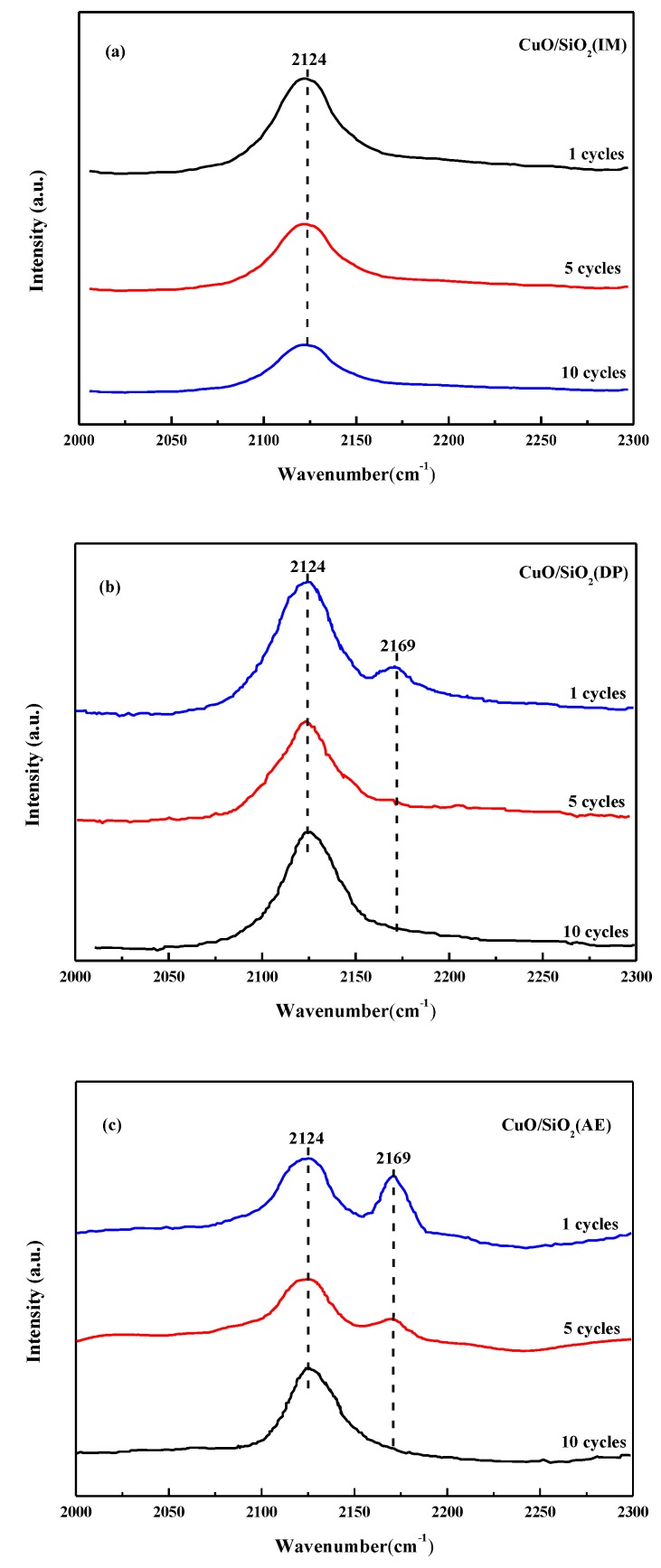
CO-IR spectra of the used CuO/SiO_2_ catalysts. (**a**) CuO/SiO_2_(IM), (**b**) CuO/SiO_2_(DP), and (**c**) CuO/SiO_2_(AE).

**Figure 10 nanomaterials-09-00842-f010:**
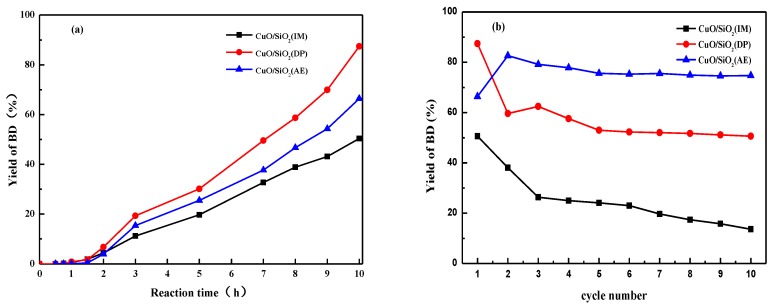
Yields of 1,4-butynediol (BD) using various CuO/SiO_2_ catalysts as functions of (**a**) reaction time and (**b**) cycle number. Reaction conditions: catalyst amount of 0.5 g, formaldehyde solution concentration of 35 vol.%, consumption of the formaldehyde solution of 50 mL, and reaction temperature of 90 °C.

**Figure 11 nanomaterials-09-00842-f011:**
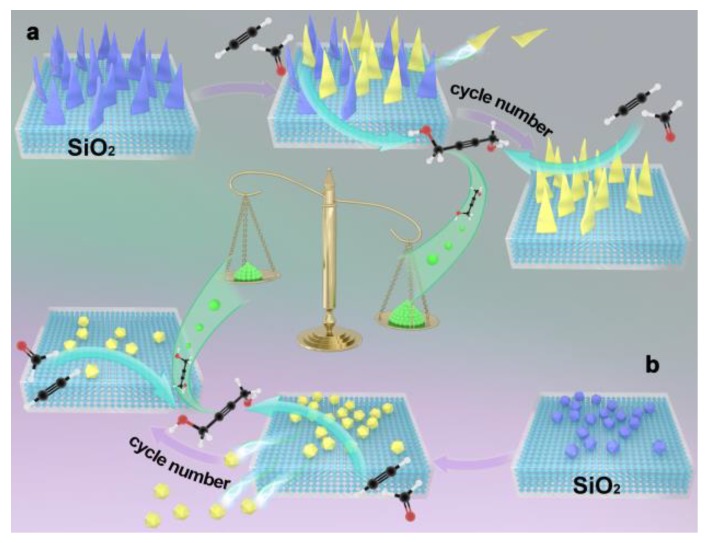
The conversion of (**a**) copper phyllosilicate and (**b**) CuO into active species and the loss of active species during the formaldehyde ethynylation.

**Table 1 nanomaterials-09-00842-t001:** Textural properties of CuO/SiO_2_ catalysts.

Catalysts	S_BET_ (m^2^·g^−1^)	D_pore_ (nm)	V_pore_ (cm^3^·g^−1^)	Grain Size *^a^* (nm)
CuO/SiO_2_(IM)	231	17.4	1.07	16.2
CuO/SiO_2_(DP)	304	7.1	0.69	7.6
CuO/SiO_2_(AE)	457	4.6	0.67	−
SiO_2_	328	9.3	0.77	−

*^a^* Calculated according to the formula D = Kλ/βcos*θ*.

**Table 2 nanomaterials-09-00842-t002:** Cu leaching of different catalysts.

Catalyst	Cu in Mother Liquid (mg/L)		
	1 Cycle	5 Cycles	10 Cycles
CuO/SiO_2_(IM)	85.3	87.6	89.3
CuO/SiO_2_(DP)	60.1	64.7	66.4
CuO/SiO_2_(AE)	35.8	38.2	36.5

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
