# Peer review of "Regulation of Cu Species in CuO/SiO2 and Its Structural Evolution in Ethynylation Reaction"

_nanomaterials, 2019, doi:10.3390/nano9060842_

Reviewer 1 Report

The paper by Li et al. reports on the characterization of Cu-species in Cu/SiO2 catalysts for the ethynylation of formaldehyde prepared by three synthesis methods (impregnation, deposition precipitation, ammonia evaporation). The nature and dispersion of Cu-species is properly assessed by different experimental methods and correlated with yield and stability of the catalysts for the target reaction.

The experimental work is properly conducted and the conclusions are consistently drawn. I believe that the paper may be suitable for publication in Nanomaterials. However, an important linguistic revision and certain clarifications about methods and results are required, as listed here below.

1)      I have to highlight important problems on the linguistic ground. A thorough proof-reading and linguistic editing is highly recommended. Beside several typos and grammar mistakes, there are often unclear sentences and terms/constructions used in a rather odd way. Few examples follow, but still, I recommend to carefully check and improve the whole paper text:

-          Abstract: “in the primary position during the reaction” -> “at the beginning of the reaction”; “slightly poor” -> “not optimal”, “limited”.

-          Page 2 line 67: “Different Cu-species existence state”-> “Different Cu-speciation” or simply “Different Cu-species”.

-          Page 2 line 82: “After aging at ambient temperature for 2 h,” in catalysis “aging” is often used with a different meaning. Here, I suggest to use “after keeping the mixture at ambient temperature for 2 h”

-          FTIR section: “vibrating”-> “vibrational”.

-          Page 7, line 225: “physical adsorbed water” -> “physisorbed water”.

-          Page 11, line 325: “Acetylene is excessive in the ethynylation of formaldehyde reaction”-> “An excess of acetylene is employed in the ethynylation of formaldehyde reaction”.

2)      About textural properties evaluation, could the authors report the N2 adsorption-desorption isotherms for the SiO2 support alone, prior to incorporation of Cu?  This will represent an important reference, to be able to state that for DP and AE methods the support pore structure is strongly perturbed (or even “destroyed” as stated at page 4, line 152).

3)      TEM: Honestly, I am not able to distinguish the claimed “layered morphology” from the image reported in Figure 3c for CuO/SiO2(AE). It seems to me very similar to Figure 3b, for the CuO/SiO2(DP). Please select a better image, or add some markers in the figure to highlight the differences discussed in the text.

4)      About Cu LMM XAES spectra in Figure 8: in all the spectra, deconvolution includes a broad component, which seems to serve as a background-like function. Which is the physical origin of this component? Moreover, it would be useful to use the same Binding Energy range for both panels (after two and six reaction cycles) and possibly a common colour code for the experimental spectra and the different deconvoluted contributions (Cu+, Cu2+, “background” contribution).

5)      CO-IR spectra: please specify (eventually in experimental) the working temperature for CO adsorption experiments.

6)      I was not able to find neither in the experimental section nor in the results section a clear definition of what is meant with “reaction cycle”. Do you run for 10 h, then stop the reaction, recover the catalyst, and start a new 10 h run? Please describe in more details the employed procedure for multi-cycle experiments.

Author Response

Response to Reviewer 1 Comments

Dear professor,

 Thank you very much for your letter and your comments on our paper "Regulation of Cu species in CuO/SiO2 and its structural evolution in ethynylation reaction". We learned a lot from the reviewers' comments, which were fair, encouraging and constructive. After carefully studying your opinions and Suggestions, we have made corresponding modifications. The main modifications are as follows:

Point 1: I have to highlight important problems on the linguistic ground. A thorough proof-reading and linguistic editing is highly recommended. Beside several typos and grammar mistakes, there are often unclear sentences and terms/constructions used in a rather odd way. Few examples follow, but still, I recommend to carefully check and improve the whole paper text:

-          Abstract: “in the primary position during the reaction” -> “at the beginning of the reaction”; “slightly poor” -> “not optimal”, “limited”.

-          Page 2 line 67: “Different Cu-species existence state”-> “Different Cu-speciation” or simply “Different Cu-species”.

-          Page 2 line 82: “After aging at ambient temperature for 2 h,” in catalysis “aging” is often used with a different meaning. Here, I suggest to use “after keeping the mixture at ambient temperature for 2 h”

-          FTIR section: “vibrating”-> “vibrational”.

-          Page 7, line 225: “physical adsorbed water” -> “physisorbed water”.

-          Page 11, line 325: “Acetylene is excessive in the ethynylation of formaldehyde reaction”-> “An excess of acetylene is employed in the ethynylation of formaldehyde reaction”.

Response 1: For the expressions and grammatical errors listed by the reviewers, we have modified them in the text part. According to the suggestion of the reviewer, we had the MDPI professional editorial team to polish and modify the full manuscript again.

Point 2:About textural properties evaluation, could the authors report the N2 adsorption-desorption isotherms for the SiO2 support alone, prior to incorporation of Cu?  This will represent an important reference, to be able to state that for DP and AE methods the support pore structure is strongly perturbed (or even “destroyed” as stated at page 4, line 152).

Response 2: According to suggestions of reviewers, we added adsorption–desorption isotherm and the pore size distribution curve of SiO2 in figure 1, and the corresponding data are shown in table 1. The analysis sections are listed in lines 155-160.

Point 3: TEM: Honestly, I am not able to distinguish the claimed “layered morphology” from the image reported in Figure 3c for CuO/SiO2(AE). It seems to me very similar to Figure 3b, for the CuO/SiO2(DP). Please select a better image, or add some markers in the figure to highlight the differences discussed in the text.

Response 3: According to suggestions of reviewers, we conducted TEM characterization of AE samples again and replaced the original figure 3c with a new clear TEM diagram.

Point 4: About Cu LMM XAES spectra in Figure 8: in all the spectra, deconvolution includes a broad component, which seems to serve as a background-like function. Which is the physical origin of this component? Moreover, it would be useful to use the same Binding Energy range for both panels (after two and six reaction cycles) and possibly a common colour code for the experimental spectra and the different deconvoluted contributions (Cu+, Cu2+, “background” contribution).

Response 4: Due to the testing conditions and objective reasons of Cu auger spectrogram, the baseline of the spectrogram is uneven, so we use spline in interpoltion method of oringe software to select the baseline. According to Suggestions of reviewers, dotted lines were used to represent the baseline and the line after peak fitting, and red lines were used to represent Cu+, green lines were used to represent Cu2+, blue lines were used to represent the fitting line, and black lines were used to represent the baseline.

Point 5:  CO-IR spectra: please specify (eventually in experimental) the working temperature for CO adsorption experiments.

Response 5: CO-IR test temperature is 25 ℃. According to the comments of reviewers, we added the test details in lines 125-127.

Point 6: I was not able to find neither in the experimental section nor in the results section a clear definition of what is meant with “reaction cycle”. Do you run for 10 h, then stop the reaction, recover the catalyst, and start a new 10 h run? Please describe in more details the employed procedure for multi-cycle experiments.

Response 6: According to the comments of reviewers, we have described the details of the cyclic experiment as follow: Cyclic experiments were carried out to determine the stability of the catalyst. For each cycle, the used catalyst running for 10 h was seperated with the reaction solution and washed with deionized water, and then start a new 10 h run.. We have added them in lines139-141.

Reviewer 2 Report

Accept in present form

Author Response

Thank you for your recognition of our work.

Reviewer 3 Report

The manuscript submitted by Y ZHAO et coll. reports on very attractive and promising copper-based nanocatalysts for performing ethynylation of formaldehyde, a process of high industrial relevance. The synthesis and the characterization of the reported CuO/SiO2 nanocatalysts are well described using adequate and conclusive analytical techniques. In terms of catalytic performance for performing ethynylation of formaldehyde, the catalysts proved to be very promising, excepted the initial CuII to CuI reduction rate that would need to be improved in the future.

In my opinion, this manuscript deserves to be published in NANOMATERIALS, after improvement of the english style and language.

Author Response

Dear professor,

 Thank you very much for your letter and your comments on our paper "Regulation of Cu species in CuO/SiO2 and its structural evolution in ethynylation reaction". We learned a lot from the reviewers' comments, which were fair, encouraging and constructive. After carefully studying your opinions and Suggestions, we have made corresponding modifications. The main modifications are as follows:

Point : In my opinion, this manuscript deserves to be published in NANOMATERIALS, after improvement of the english style and language.

         Response : According to the suggestion of the reviewer, we had the MDPI professional editorial team to polish and modify the full manuscript again.

Reviewer 4 Report

The paper dealing with different preparation methods of CuO/SiO2 materials is interesting and overall well written. 

Nevertheless I think that there are few points to be addressed before its publication.

1) textural properties: I think that it would be interesting to have informations about the bare silica, in order to better evaluate the changes occurring upon copper deposition. 

2) TEM characterization.: the authors state that. while CuO/SiO2(DP) presents a large number of spherical CuO particles, the AE sample shows a layered morphology. I do not see many differences between the photos corresponding to this two samples.Fig.3 B and C seem to be very similar.

3) XPS and Auger characterization: I do not understand why the XPS characterization has been made only in the fresh samples while the auger only in the spent samples. A comparison between fresh and spent samples with both techniques, would be more interesting. Moreover, it would be interesting to have the XPS derived atomic ratio between Cu and Si before and after reaction to get clues about the the dispersion differences and the phenomena occurring during reaction

4) The authors should do a quantitative analysis of the spent samples or of the solution to confirm and quantify the leaching of copper they hypothesize during reaction                                                                                                                                                                                                                                                                                                                                                         

Author Response

Dear professor,

    Thank you very much for your letter and your comments on our paper "Regulation of Cu species in CuO/SiO2 and its structural evolution in ethynylation reaction". We learned a lot from the reviewers' comments, which were fair, encouraging and constructive. After carefully studying your opinions and Suggestions, we have made corresponding modifications. The main modifications are as follows:

Point 1: textural properties: I think that it would be interesting to have informations about the bare silica, in order to better evaluate the changes occurring upon copper deposition. 

Response 1:According to Suggestions of reviewers, we added adsorption–desorption isotherm and the pore size distribution curve of SiO2 in figure 1, and the corresponding data are shown in table 1. The analysis sections are listed in lines 155-160.

Point 2: TEM characterization.: the authors state that. while CuO/SiO2(DP) presents a large number of spherical CuO particles, the AE sample shows a layered morphology. I do not see many differences between the photos corresponding to this two samples.Fig.3 B and C seem to be very similar. 

Response 2: According to Suggestions of reviewers, we conducted TEM characterization of AE samples again and replaced the original figure 3c with a new clear TEM diagram.

Point 3: XPS and Auger characterization: I do not understand why the XPS characterization has been made only in the fresh samples while the auger only in the spent samples. A comparison between fresh and spent samples with both techniques, would be more interesting. Moreover, it would be interesting to have the XPS derived atomic ratio between Cu and Si before and after reaction to get clues about the the dispersion differences and the phenomena occurring during reaction.

Response 3: The Cu valence state in fresh sample was Cu2+, and the valence state of Cu could be explained by XPS analysis alone. The valence state of Cu species may change from Cu2+ to Cu+ and Cu0 under the atmosphere of formaldehyde-acetylene reduction. It is difficult to distinguish Cu+ and Cu0 by XPS analysis alone, while XAES has obvious differences on Cu+ and Cu0. Therefore, we conducted XPS analysis on the samples before the reaction and XAES analysis on the samples after the reaction. As the reviewer stated, the data of Cu/Si ratio on the catalyst surface can be obtained by XPS. But, because of the particularity of the acetylation reaction, CuO species will activate in situ to cuprous acetylene in the catalytic reaction. In this process, acetylene C atoms replace O in CuO, which complicates the surface atoms of the catalyst. It is difficult to predict the phenomena encountered in the reaction and dispersion by atomic ratio. Therefore, the data of Cu/Si ratio are not listed in this paper.

Point 4: The authors should do a quantitative analysis of the spent samples or of the solution to confirm and quantify the leaching of copper they hypothesize during reaction.

Response 4: According to Suggestions of reviewers, ICP analysis of copper content in the reaction solution preserved in previous experiments was carried out. The analytical data and corresponding descriptions are listed in lines 335-338. ICP test methods and conditions are listed in lines 128-130.

Round  2

Reviewer 4 Report

I think that authors answered to all comments.